# Identifying Soccer Players’ Playing Styles: A Systematic Review

**DOI:** 10.3390/jfmk8030104

**Published:** 2023-07-26

**Authors:** Spyridon Plakias, Serafeim Moustakidis, Christos Kokkotis, Marina Papalexi, Themistoklis Tsatalas, Giannis Giakas, Dimitrios Tsaopoulos

**Affiliations:** 1Department of Physical Education and Sport Science, University of Thessaly, 38221 Trikala, Greece; spyros_plakias@yahoo.gr (S.P.); ttsatalas@uth.gr (T.T.); ggiakas@gmail.com (G.G.); 2AIDEAS OÜ, Narva mnt 5, 10117 Tallinn, Estonia; 3Department of Physical Education and Sport Science, Democritus University of Thrace, 69100 Komotini, Greece; ckokkoti@affil.duth.gr; 4Department of Operations, Technology, Events and Hospitality Management, Manchester Metropolitan University, Oxford Road, Manchester M15 6BH, UK; m.papalexi@mmu.ac.uk; 5Center for Research and Technology Hellas, Institute for Bio-Economy & Agri-Technology, 60361 Volos, Greece; d.tsaopoulos@certh.gr

**Keywords:** footballers, game style, soccer tactics, performance analysis, player positions

## Abstract

Identifying playing styles in football is highly valuable for achieving effective performance analysis. While there is extensive research on team styles, studies on individual player styles are still in their early stages. Thus, the aim of this systematic review was to provide a comprehensive overview of the existing literature on player styles and identify research areas required for further development, offering new directions for future research. Following the PRISMA guidelines for systematic reviews, we conducted a search using a specific strategy across four databases (PubMed, Scopus, Web of Science, and SPORTDiscus). Inclusion and exclusion criteria were applied to the initial search results, ultimately identifying twelve studies suitable for inclusion in this review. Through thematic analysis and qualitative evaluation of these studies, several key findings emerged: (a) a lack of a structured theoretical framework for player styles based on their positions within the team formation, (b) absence of studies investigating the influence of contextual variables on player styles, (c) methodological deficiencies observed in the reviewed studies, and (d) disparity in the objectives of sports science and data science studies. By identifying these gaps in the literature and presenting a structured framework for player styles (based on the compilation of all reported styles from the reviewed studies), this review aims to assist team stakeholders and provide guidance for future research endeavors.

## 1. Introduction

Performance analysis in football is not only a crucial tool for coaches [1,2], but also a subject of extensive research in recent years [3,4]. The advancement of technology has provided team analysts and researchers with a vast array of data [5]. While this facilitates the work of analysts and researchers, it also raises the question of expediting analysis without compromising its effectiveness [6].

Traditionally, performance analysis has emphasized the use of isolated performance indicators that provide valuable information to coaching staffs of teams [7]. However, in recent years, there has been a growing emphasis on playing styles among scientists, as they believe that playing styles can better explain football, particularly in terms of tactical aspects [8,9]. Playing style is generally defined as the characteristic playing patterns adopted by a team [10]. The playing styles of teams have been researched for a long time [11]. In contrast, research on the playing styles of individual players has emerged more recently, as previous years primarily focused on evaluating and rating players based on performance indicators [12,13,14]. However, the performance of soccer players is multifactorial in nature, influenced by factors such as technique, fitness, individual tactics, psychology, personality, and physical characteristics [15,16,17,18] in combination with contextual variables like match status and location. Therefore, by considering all these factors, the individual profile-style of each football player can be determined. As a result, soccer players can adopt different styles even when playing in the same position, highlighting that traditional positional labels (goalkeeper, defender, midfielder, forward) are no longer sufficient [19].

A definition of players’ playing styles has been given by Decroos and Davis [20]. According to them, a player’s playing style can be described by examining their preferred position on the field and the types of actions and movements they tend to execute during a game. Only a few classical inductive statistical methods can meet the specific requirements of grouping such features, such as principal components analysis and k-means clustering. In contrast, there are numerous artificial intelligence (AI) techniques that can offer solutions to this particular issue [21,22]. Additionally, AI methods can effectively handle the management of “big data” [23,24,25]. These two reasons have led to the widespread utilization of AI methods in game style recognition in recent years [22].

Understanding the playing style of soccer players can provide coaches with valuable insights in many areas: (1) Player scouting [26,27]—Teams require players whose style aligns with the team’s play style and the other players. Otherwise, they may invest a significant amount of money in a player who, despite their individual talent, cannot effectively contribute to the team. (2) Player development [18,20]—Knowledge of a player’s playing style can be utilized to assist them in improving their skills and game. By identifying areas where a player may need improvement, coaches can tailor training and development programs to address weaknesses and enhance the player’s strengths. (3) Team selection and tactics [19,28]—Coaches can utilize knowledge of a player’s playing style to select the most suitable players for a particular match. For instance, if a coach intends to adopt a counter-attacking strategy, they will prioritize fast strikers who can excel in that style. (4) Opponent analysis [12,29]—Identifying the playing style of opponents can assist teams in preparing more effectively for matches. By studying the playing style of opposing players, coaches can anticipate their tactics and adjust their own game plan accordingly.

Although team playing styles have recently been reviewed in the literature [30], no comprehensive review of existing research on individual soccer players’ playing styles has been conducted to date. Furthermore, given that the specific topic has only recently gained significant research attention, it is important to highlight the gaps that exist in the literature, both theoretically and methodologically. Therefore, considering the practical value of recognizing players’ styles (as discussed in the previous paragraph), the aim of this study is to systematically review the existing studies on this specific subject in order to: (a) provide a comprehensive overview of the current knowledge to assist coaches and team performance analysts, and (b) assess the existing literature, identify its gaps, and propose new directions for future research.

## 2. Materials and Methods

### 2.1. Reporting

The review was conducted based on the PRISMA guidelines [31,32,33]. It was not approved for registering because “PROSPERO does not accept reviews of sporting performance and the specific review does not include a direct health-related outcome”. Ethical approval is not required because this review will use already-published data that are not related to either human or animal use [34,35,36].

### 2.2. Literature Search

Two authors (S.P. and C.K.) conducted independent searches on 21 April 2023, using four databases (PubMed, Scopus, Web of Science, and SPORTDiscus). In cases where discrepancies arose between the two authors, a third author (S.M.), who has experience in conducting systematic reviews, made the final decision regarding the inclusion or exclusion of the articles in question. The search strategy applied for all databases was as follows: ((“soccer player”) OR (footballer)) AND ((“playing style”) OR (“play style”) OR (“game style”) OR (“playing role”)). There were no restrictions on the publication date of the studies.

The selection criteria for the articles involved two stages. In the first stage (screening), the titles and abstracts were considered, while in the second stage (eligibility), the full text of the articles was reviewed. The inclusion and exclusion criteria used during these stages are presented in Table 1. Furthermore, additional papers were identified through the references of the already-selected studies, using a technique known as the “snowball effect”.

### 2.3. Data Extraction

A Microsoft Excel spreadsheet was used for recording the retrieved data. Title, author name, publication year, aim, type and size of sample, method of data analysis, and results were recorded for each retrieved paper.

### 2.4. Quality Assessment

All of the included studies were evaluated using the answers to ten questions (Q1–10). For this purpose, a two-point scale was adopted (where “yes” = 1 point; “no” = 0). An 11th question (Q11) concerned the publication of the article in a journal or conference (Table 2). The checklist developed ad hoc [37,38], taking into account questions that had been used in previous systematic reviews in the field of sports [37,39] and the aims of our review. The agreement between the two authors was measured by Cohen’s kappa coefficient (k = 0.98). Furthermore, the points from all ten questions were added up for each study. The rating varied from 0 to 10 points. Quality assessment was not applied for the purpose of inclusion of the studies in the review. Finally, the averages were calculated for the studies that had been published in a journal and those that had been presented at conferences.

## 3. Results

### 3.1. Search, Selection, and Inclusion of Publications

The initial search of the four databases yielded 646 titles. After removing duplicates, 391 articles were screened for relevance on the basis of their title and abstract. As a result, 29 full texts were assessed for eligibility; 19 of them were considered unsuitable for the review, but from the references of the 10 articles that were considered suitable, another 2 investigations were found. Therefore, a total of 12 articles were included in the review. The reasons for excluding the remaining articles were: review articles (N = 44), articles related to teams’ playing styles (N = 54), other sports (N = 36), academies (N = 26), women (N = 4) or small-side-games (N = 14), books (N = 2), dissertations (N = 6), other types of non-peer-reviewed documents (Ν = 12), non-English language articles (N = 4). Additionally, 170 articles were excluded because of their purpose and 5 because the full text was not available (Figure 1).

### 3.2. Quality Assessment

As can be seen from Table 3, one study was evaluated on the 10-point scale with 4, three studies with 5, two studies with 6, three with 7, one with 8, and two with 9. Four studies were published in journals, while eight in conferences. The average score for all studies was 6.33, for journal studies was 7.25, while for conference papers it was 5.88.

### 3.3. Descriptive Analysis

Table 4 presents the breakdown of the 12 studies, indicating that 9 of them employed artificial intelligence techniques such as machine learning and deep learning. The remaining 3 studies utilized alternative methods, including spatiotemporal qualitative calculus, statistics, and computational approaches. It is noteworthy that all the studies were published within the past 4 years, except for one study published in 2004. Figure 2 illustrates the distribution of studies per year, categorized based on their utilization of artificial intelligence.

Regarding data sources, websites and analysis platforms played a prominent role. Specifically, the Whoscored website was utilized four times, the FBref website once, the Wyscout platform twice, and the Statsbomb platform once. Furthermore, GPS devices, camera-based optical tracking data, Pappalardo’s public dataset, and the electronic game “EA Sports FIFA” were each employed in one study. In one instance, the events were manually coded by the first author, while in another study, the data source was not explicitly mentioned.

The data employed in the studies encompassed various types. Tracking data were utilized in 3 studies, event stream data were utilized in 9 out of the 12 studies, 2 studies reported using match sheet data, and 1 study relied on player evaluation attributes extracted from a video game. Among the studies, 8 had access to data from over 750 matches, 2 studies used a limited number of matches (1 and 22, respectively), and 2 studies did not specify the number of matches from which their data were collected.

## 4. Discussion

### 4.1. Quality Assessment

The qualitative evaluation of the studies reveals that researchers have yet to address (0%) the impact of contextual variables (e.g., match status, match location, type of competition, or quality of the opponent) on players’ playing style. It has been established that contextual variables influence team playing styles [48,49,50,51,52,53,54]. Therefore, it would be highly valuable to investigate how these variables also affect the style of individual players.

Moreover, deficiencies were identified in other aspects of research quality. Firstly, the mention or measurement of reliability/validity of the data provider was very low (25%) among the studies. In particular, in the studies of Beernaerts, De Baets [41] and Li, Zong [12], the validity is stated, while in the study of Taylor, Mellalieu [45], the validity is measured. Secondly, in three studies [40,41,42], their purpose was not clearly stated. Finally, in three studies [45,46,47], the characteristics of the sample (and the data) are not completely clear.

On the other hand, other quality criteria yielded high percentages. Specifically, all studies had relevant background literature, and the conclusions drawn were supported by the results. Additionally, only one study [21] lacked clear presentation of the results.

### 4.2. Theoretical Framework for Players’ Playing Styles

According to Goes, Meerhoff [6], many computer science studies neglect to provide theoretical context to explain tactical behavior. This issue is also prevalent in most of the studies included in this review. Particularly, five studies [20,21,29,40,42] compare players based on vectors without describing any specific playing styles associated with different positions in the team formation. Similarly, two studies [41,47] that employed other computational methods did not provide descriptions of player styles. This does not necessarily imply that these investigations lack quality, but it suggests that they may have different objectives or purposes.

At the same time, two studies that utilized supervised machine learning methods employed pre-existing frameworks for categorizing players’ playing styles. Specifically, Aalbers and Van Haaren [43] proposed a framework consisting of 21 player styles after conducting searches in sports media and the video game Football Manager 2018. On the other hand, Ghar, Patil [46] introduced a framework comprising 15 player styles but did not specify the source(s) they relied on for its development.

In contrast, three papers not only aimed to identify styles of play but also sought to describe the playing styles adopted by players in different positions within the team formation. Firstly, Taylor, Mellalieu [45] recognized variations in playing styles among players in the same position but could not generalize the results due to the small sample size of only 22 soccer players. Nonetheless, their study provided valuable insights for future researchers to explore distinct playing styles adopted by soccer players. Subsequently, Lee, Li [44] discovered 18 play styles, while Li, Zong [12] discovered 14 player styles (18 when including styles that were similar for both the right and left sides of the field). It is worth noting that the player styles identified in the latter two studies may be limited by their reliance solely on match event data. The inclusion of fitness performance indicators or more detailed tracking data could provide additional information on players’ styles.

To address the gap in the existing literature, we synthesized all player styles mentioned in the 12 studies included in this review and developed a framework consisting of 26 player styles (33 when including styles that were similar for both sides of the field) (Figure 3). Furthermore, Table 5 provides definitions and examples of players for each style. In formulating the definitions, we considered sources beyond those presented in this review, while the examples are based on the personal perception of the first author (S.P.).

Although we have established a useful framework for future researchers and soccer practitioners, it is essential to acknowledge certain limitations. Firstly, the review’s scope was quite narrow, as it solely focused on research articles published in scientific journals or conferences, excluding other valuable sources such as reviews and books. Additionally, the investigations were limited to 11 × 11 soccer matches, specifically involving adult professional male soccer players, omitting women’s and development soccer. Nonetheless, these criteria were necessary to ensure the conclusions drawn align with the reality of the professional competitive level. On the other hand, while we conducted a qualitative evaluation of the included articles, it did not influence their inclusion in the review. Our decision was intentional, aiming to avoid further reducing the already limited number of studies and to encompass well-established player styles utilized in everyday soccer practice. Despite some articles merely mentioning player styles without in-depth explanations of their role on the field, we incorporated styles in developing the framework to ensure its comprehensiveness.

## 5. Conclusions

Through this review, we aimed to summarize the existing scientific knowledge regarding the playing styles of soccer players. Despite the limitations of our study, we have found that recognizing different player styles can have practical implications in various areas, including player scouting, player development, team selection and tactics, and opponent analysis. Additionally, we addressed a gap in the current literature by developing a comprehensive framework that encompasses all identified player styles for each position. This framework can be valuable for future researchers and soccer professionals, including coaches, scouts, analysts, and technical directors. Moreover, our qualitative assessment of the included studies revealed several methodological deficiencies, and we also observed a lack of research examining the influence of contextual variables on players’ styles. Lastly, we emphasized the need for collaboration between sports science and data science to bridge the gap between theoretical knowledge and practical applications. Based on our findings and considering the absence of previous reviews in this rapidly growing field of research, we believe that this systematic review will open new avenues in soccer performance analysis.

## Figures and Tables

**Figure 1 jfmk-08-00104-f001:**
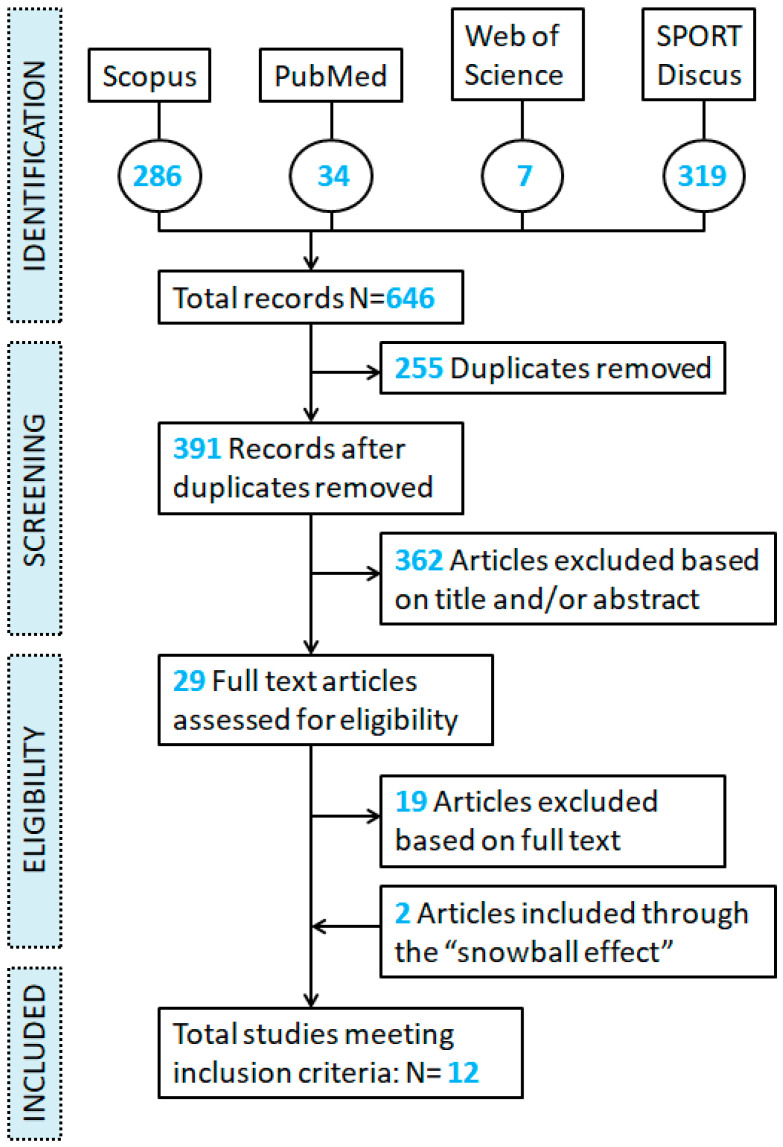
Flow chart of the literature search according to PRISMA guidelines.

**Figure 2 jfmk-08-00104-f002:**
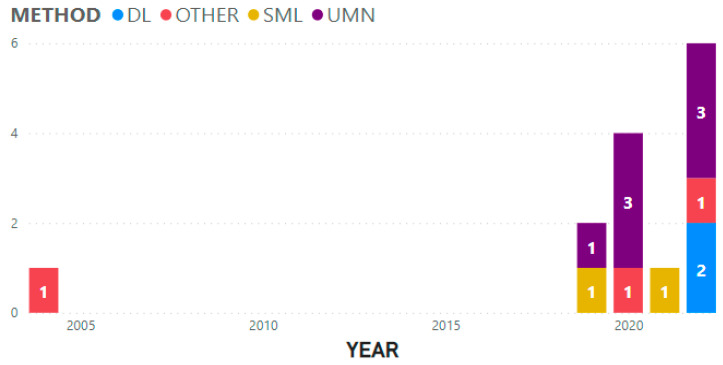
Types of methods by year. DL = Deep learning, SML = Supervised machine learning, UMN = Unsupervised machine learning, OTHER = Other methods. Studies that used more than one of the types of methods considered in the chart are cited more than once (as many times as the types of methods used). The numbers in the diagram indicate the number of studies in each method type.

**Figure 3 jfmk-08-00104-f003:**
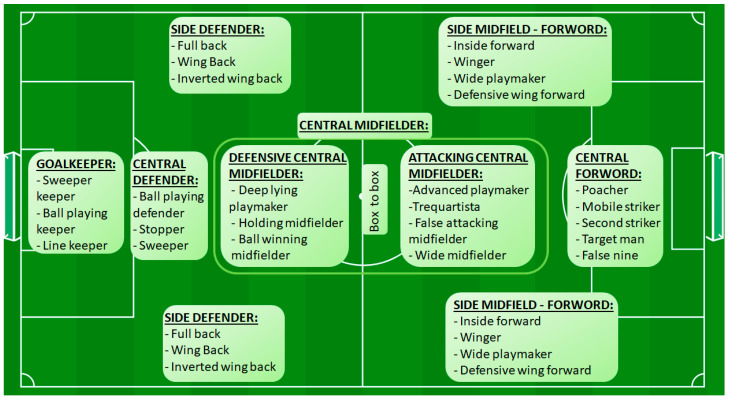
Synthesis of player styles reported in the 12 studies included in this review.

**Table 1 jfmk-08-00104-t001:** Inclusion and exclusion criteria applied during the selection stages (screening and eligibility).

Inclusion Criteria	Exclusion Criteria
Original article	Unavailable full text
English full text	Full text in a language other than English
Published in peer-reviewed journal or conference	Review articles, editorials, commentaries, dissertations, and book chapters
The terms reported in the search strategy should be mentioned in the title, abstract, or body of the text	Articles related to other sports, robotic soccer, and women’s soccer
At least one of the purposes of the article should be about identifying the playing style of soccer players	Articles sampled were about developmental ages and not high-level professional soccer
	Articles that the data were taken from small-side-games and not from the regular 11 vs. 11 game.
	Articles that dealt with the playing styles of teams rather than individual players.

**Table 2 jfmk-08-00104-t002:** Questions for the quality assessment of the included studies.

Q1	The study objective(s) is/are clearly set out
Q2	Relevance of background literature
Q3	The characteristics of the sample are clearly defined (number of players, number of matches, number of observations)
Q4	Variables apply to all phases of the game
Q5	The reliability/validity of the data provider is stated, is mentioned, or is measured
Q6	Certain contextual variables (e.g., match status, match location, type of competition, or quality of the opponent) are taken into account
Q7	The results are clearly presented
Q8	Specific player styles were recognized with clearly refined characteristics for each of them
Q9	A distinction is made according to player positions
Q10	Conclusion supported by results
Q11	Journal/Conference

**Table 3 jfmk-08-00104-t003:** The quality assessment scores for each study.

Article	Q1	Q2	Q3	Q4	Q5	Q6	Q7	Q8	Q9	Q10	T	Q11
[12]	1	1	1	1	1	0	1	1	1	1	9	J
[20]	1	1	1	0	0	0	1	0	0	1	5	C
[21]	1	1	1	0	0	0	1	0	0	1	5	J
[29]	1	1	1	1	0	0	1	0	0	1	6	C
[40]	0	1	1	1	0	0	1	0	0	1	5	C
[41]	0	1	1	1	1	0	1	0	0	1	6	J
[42]	0	1	1	1	0	0	0	0	0	1	4	C
[43]	1	1	1	1	0	0	1	1	1	1	8	C
[44]	1	1	0	1	0	0	1	1	1	1	7	C
[45]	1	1	1	1	1	0	1	1	1	1	9	J
[46]	1	1	0	1	0	0	1	1	1	1	7	C
[47]	1	1	0	1	0	0	1	0	0	1	5	C
Summary percent	75	100	75	83	25	0	92	42	42	100	63	

T = Total score for each study, Summary percent = Sum of scores of all studies on each question (in percentage %), C = Conference, J = Journal.

**Table 4 jfmk-08-00104-t004:** Methodologies, data, and output of the studies.

Title	Methods	Category of Method	Data Source	Kind of Data	Games/Competitions	Output
[12]	k-means algorithm	Unsupervised machine learning	Whoscored	Match event data with spatial information	960 matches of 2016–2019 Chinese CSL	Discovered and named 18 distinct playing styles
[20]	Manhattan distance, Euclidean distance	Unsupervised machine learning	Whoscored	Match sheet and event data with spatial information	9155 matches (English Premier League, German Bundesliga, Spanish Primera Division, Italian Serie A and French Ligue One/2012/13 to 2016/17)	Summarize the playing style in a fixed-length player vector
[21]	Convolutional Autoencoder, Euclidean distance, Cosine distance, and Manhattan distance	Deep learning and clustering	Pappalardo’s public dataset (initially obtained from Wyscout)	Ball events data with information of involved players, position, time, outcome, and type of action	1826 games (first divisions in England, France, Germany, Italy, and Spain from the 2017/2018 season)	Construct a player’s passing style descriptor
[29]	Mixture models	Unsupervised machine learning	Statsbomb	Event stream data with temporal & spatial information	760 games of English Premier League (2017/18 and 2018/19)	SoccerMix, which partitioning player actions into groups of similar actions, constructs a vector that describes a specific player’s style
[40]	K-means clustering, 3 identical subnetworks (sharing weights) named 2MapNet, each of which has two branch convolutional neural networks (CNNs)	Unsupervised machine learning and deep learning	GPS devices	Tracking data	750 matches of 2019 and 2020 South Korean K League 1 and 2	A model (6MapNet) to capture players’ playing style
[41]	Qualitative Trajectory Calculus (QTC)	Spatiotemporal qualitative calculus	Camera-based optical tracking	Tracking data	1 match of a 2016–2017 professional soccer	Characterization of playing styles by describing the movement behavior of players
[42]	Unsupervised clustering based on Euclidean distance	Unsupervised machine learning	Whoscored	Match sheet and event data with temporal & spatial information	9155 matches (English Premier League, German Bundesliga, Spanish Primera Division, Italian Serie A and French Ligue One/2012/13 to 2016/17)	A player vector that characterizes the playing style of a player by simply concatenating his feature vectors for each event type
[43]	Stochastic Gradient Descent classifier	Supervised machine learning	Wyscout	Event stream data, with a reference to: team, player, type of event, start location on the pitch (and when relevant also the end location)	2017/2018 season of the English Premier League, Spanish LaLiga, German 1 Bundesliga, Italian Serie A, French Ligue One, Dutch Eredivisie and Belgian Pro League	Identification of the most suitable players for the central midfielder roles
[44]	Boruta algorithm, K-means clustering	Feature selection and unsupervised machine learning	Whoscored	Performance indicators	1900 games of English Premier League (2014/15–2018/19)	Discovered and named 18 distinct playing styles
[45]	Chi-square	Statistics	Coded by the first author using computerised system	Performance indicators with their outcomes	22 matches played by a professional British soccer team in both cup and league competition during the 2002–03 domestic season	Individual profiles in the same playing position exhibit differences
[46]	Logistic regression and random forest regression	Supervised machine learning	EA Sports FIFA, FBref, Wyscout	Attributes and events	Not mentioned	An automated scouting system, i.e., an algorithm to suggest suitable players according to the requirements of the manager
[47]	Graph-based change-point detection (CPD) algorithm named discrete g-segmentation, Clustering Based on Mean Role-Adjacency Matrices	Computational methods	Not mentioned	Spatiotemporal Tracking Data	Not mentioned	A framework (SoccerCPD) that distinguishes tactically intended formation and role changes from temporary switches in soccer matches

**Table 5 jfmk-08-00104-t005:** Definitions and examples of player styles listed in Figure 3.

Style	Definition	Examples
Sweeper keeper	Is a goalkeeper who frequently comes off their goal line to act as an additional defender. They are known for their ability to read the game well, anticipate attacks, and make sweeping clearances outside of their penalty area [46,55].	Manuel Neuer, Ederson Moraes
Ball-playing keeper	Is comfortable with the ball at their feet and is actively involved in the team’s build-up play. They possess good passing, dribbling, and decision-making skills, often contributing to the team’s possession-based style of play [43,44].	Ter Stegen, De Gea
Line keeper	Also known as shot-stopper or classic goalkeeper, focuses primarily on making saves and protecting their goal. They tend to stay closer to their goal line and excel in shot-stopping abilities, including reflexes, positioning, and aerial dominance [44,46].	Gianluigi Buffon, Keylor Navas
Ball-playing defender	Is a central defender who excels in distributing the ball, initiating attacks from the back, and participating in the team’s build-up play. They are comfortable on the ball, have good passing ability, and often contribute to the team’s possession-based style of play [12,46].	Sergio Ramos, Gerard Piqué
Stopper	Is a central defender who primarily focuses on defensive duties and is known for their physicality, strong tackling, and aerial prowess. They excel in winning duels, intercepting passes, and providing a solid defensive presence to protect the goal [12,46].	Kalidou Koulibaly, Giorgio Chiellini
Sweeper	Or libero is a central defender who operates as the last line of defense, positioned behind the main defensive line. They have excellent reading of the game, anticipation skills, and are responsible for clearing any potential threats that breach the defensive line [56,57].	Franco Baresi, Traianos Dellas
Full back (defensive)	Is a player positioned on the defensive line whose primary responsibilities include defensive duties, marking opposing wingers, and providing support in both defensive and offensive phases of play [12,46].	César Azpilicueta, Dani Carvajal
Wing back (offensive)	Is a player who operates as a hybrid between a full back and a winger. They have defensive responsibilities but are also encouraged to push forward and provide width in the attacking third. Wing backs often contribute to the team’s attacking play and are known for their pace and crossing ability [12,46].	Achraf Hakimi, Joshua Kimmich
Inverted wing back	Is a player who operates as a full back or wing back but prefers to cut inside and contribute to the central areas of the pitch rather than hugging the touchline. They provide a different dynamic to the team’s attack by creating overloads and offering passing options in central areas [58,59].	Philipp Lahm, João Cancelo
Deep lying playmaker	Is a midfielder who operates in a deeper position on the field, typically just in front of the defensive line. They are responsible for dictating the team’s play from a deeper position, distributing accurate long-range passes, and initiating attacks from the defensive third [43,44].	Andrea Pirlo, Xabi Alonso
Holding midfielder	Or anchor, is a player whose primary role is to provide defensive stability and shield the backline. They excel in breaking up opposition attacks, intercepting passes, and disrupting the opponent’s play in the midfield area [43,44].	Sergio Busquets, Fabinho
Ball winning midfielder	Or destroyer or enforcer, is a player who specializes in regaining possession for their team by winning tackles, intercepting passes, and applying pressure on the opposition. They play a vital role in breaking down the opponent’s attacks and disrupting their rhythm [43,46].	Casemiro, Kanté
Box to box	Is a versatile player who covers a large area of the pitch, contributing both defensively and offensively. They are known for their stamina, work rate, and ability to make significant contributions in both defensive and attacking phases of play [43,46].	Jordan Henderson, Arturo Vidal
Advanced playmaker	Is a central attacking midfielder who operates in an advanced position, typically just behind the forward(s). They are creative and influential playmakers who excel in their vision, passing ability, and ability to unlock defenses with through balls and key passes [43,46].	Kevin De Bruyne, Mesut Özil
Trequartista	Is a central attacking midfielder who primarily focuses on creating goal-scoring opportunities and making incisive runs into the opposition’s penalty area. They are typically the most advanced midfielder and are known for their dribbling skills, close control, and ability to score goals [44,60].	Lionel Messi, Paulo Dybala
False attacking midfielder	Is a player who operates in a central position but often drops deeper into midfield or moves out wide to create space for other attackers. They aim to confuse the opposition’s defense and create openings by dragging defenders out of position with their movement and positional versatility [44,61].	David Silva, Isco
Wide midfielder	Or mezzala or half-winger is a central attacking midfielder who tends to move laterally, overloading the wings and creating outnumbers for his team. They show a preference for flank dribble, flank pass, flank long pass, and ball recovery in the mid-front area [12,62].	Bernardo Silva, Mathieu Valbuena
Inside forward	Or inverted winger, is a wide midfielder who operates in a more central position, cutting inside towards goal rather than staying wide. They often aim to create goal-scoring opportunities by making diagonal runs, dribbling towards the center, and shooting from inside the box [12,63].	Mohamed Salah, Angel di Maria
Winger	Is a wide midfielder whose primary role is to provide width and deliver crosses into the box from the flanks. They are known for their speed, dribbling ability, and ability to take on defenders in one-on-one situations. Wingers often look to create scoring opportunities for teammates with accurate crosses or by cutting inside to shoot themselves [12,44].	Cristiano Ronaldo, Gareth Bale
Wide playmaker	Is a creative midfielder who operates in wider areas of the pitch. They possess excellent vision, passing ability, and decision-making skills, and are responsible for creating goal-scoring opportunities by delivering accurate crosses, through balls, or making incisive passes from the flanks [64,65].	Franck Ribéry, David Beckham
Defensive wing forward	Is a wide midfielder who combines offensive and defensive responsibilities. They not only contribute to the team’s attacking play but also actively track back, press opponents, and provide defensive support to their team. They play a crucial role in maintaining defensive solidity while also providing an attacking threat [44,66].	Raheem Sterling, Dirk Kuyt
Poacher	Is a central forward who specializes in capitalizing on goal-scoring opportunities inside the penalty box. They have a natural ability to find space in the box, time their runs, and pounce on loose balls to score goals [12,46].	Robert Lewandowski, Erling Haaland
Mobile striker	Is a forward who possesses pace, agility, and good movement off the ball. They are known for their ability to make intelligent runs, exploit spaces, and contribute to the build-up play by linking with teammates [12,44].	Pierre Aubameyang, Timo Werner
Second striker	Or “supporting “ or “shadow” striker is a forward who operates just behind the main central striker. They have creative abilities and excel in providing assists, making key passes, and scoring goals by arriving late into the box [12,44].	Thomas Müller, Ciro Immobile
Target man	Is typically tall, strong, and adept at holding up the ball with their back to the goal. They often act as a focal point for the team’s attacks, using their physical presence to win aerial duels and bring teammates into play [12,44].	Olivier Giroud, Edin Džeko
False nine	Is a forward who operates in a deeper position than traditional central forwards. They drop deep into midfield to create space and disrupt the opposition’s defensive structure. They are technically skilled, good at dribbling, and have excellent vision to create scoring opportunities for themselves and teammates [47,67].	Roberto Firmino, Antoine Griezmann

## Data Availability

Data is not unavailable.

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
