# Peer review of "Identifying Soccer Players’ Playing Styles: A Systematic Review"

_jfmk, 2023, doi:10.3390/jfmk8030104_

Round 1

Reviewer 1 Report

The identification of playing styles in football is a relatively rare topic, and I appreciate the authors who attempted a systematic review in the given topic, while observing scientific methodological principles. The work would have benefited from a separate Limitations of the study paragraph, although partial limitations of the study are listed in the text. Errors should be corrected in the manuscript, Figure 3 should be indicated on line 243 and not Figure 2 (it is already on page 9, line 187). Table 5, line 244 lists Definitions and examples of playing style "listed in Figure 4.1 - but this Figure 4.1 is not listed in the manuscript. In Figure 2, there are bar graphs in shades of gray that are difficult to distinguish from each other.

The English used is correct and readable.

Author Response

We sincerely appreciate your positive feedback. In response to your suggestions, we have incorporated a separate paragraph discussing the limitations of our study at the end of the Discussion section, just before the Conclusions. Additionally, we made the corrections you pointed out: in line 243, we replaced "Figure 2" with "Figure 3," and in line 244, we replaced "Figure 4.1" with "Figure 3."

Regarding Figure 2, we would like to clarify that there are no missing bars; however, a significant gap exists in the relevant literature from 2005 to 2018. This gap occurred as researchers were not actively engaged during that period, but in 2019, they resumed their work with renewed enthusiasm, driven by the possibilities presented by advancements in technology.

Once again, we are grateful for your valuable feedback, which has contributed to the improvement of our work.

Reviewer 2 Report

At the outset, I would like to thank you for inviting me to review your work.

The article is correctly edited, the division of the content is appropriate, the problem posed was solved correctly methodically.

My comments relate only to Table 3 and Table 4.

The ordinal variable "Article" / "Title" should be arranged in ascending order.

In addition, the caption of Table 3 is missing.

Author Response

We genuinely appreciate your positive feedback. Following your suggestion, we have made the necessary adjustments. Specifically, we have sorted the "Article" / "Title" column in ascending order in Tables 3 and 4. Additionally, we have included the missing caption in Table 3.